# Food insecurity and hypertension: A systematic review and meta-analysis

Sourik Beltrán[1,2☯], Marissa Pharel[3☯], Canada T. Montgomery[1], Itzel J. López-Hinojosa[4], Daniel J. Arenas[1], Horace M. DeLisser[1]*

1 Academic Programs Office, Perelman School of Medicine, University of Pennsylvania, Philadelphia, Pennsylvania, United States of America, 2 Department of Medical Ethics and Health Policy, Perelman School of Medicine, University of Pennsylvania, Philadelphia, Pennsylvania, United States of America, 3 Rush Medical College, Rush University, Chicago, Illinois, United States of America, 4 Pritzker School of Medicine, University of Chicago, Chicago, Illinois, United States of America

☯ These authors contributed equally to this work.
* delisser@pennmedicine.upenn.edu

**Data Availability Statement:** All relevant data are within the manuscript and its Supporting Information files.

## Abstract

### Background

Food insecurity (FIS) is an important public health issue associated with cardiovascular risk. Given the association of FIS with diets of poorer nutritional quality and higher salt intake as well as chronic stress, numerous studies have explored the link between FIS and hypertension. However, no systematic review or meta-analysis has yet to integrate or analyze the existing literature.

### Methods

We performed a wide and inclusive search of peer-reviewed quantitative data exploring FIS and hypertension. A broad-terms, systematic search of the literature was conducted in PubMed, Embase, Scopus, and Web of Science for all English-language, human studies containing primary data on the relationship between FIS and hypertension. Patient population characteristics, study size, and method to explore hypertension were extracted from each study. Effect sizes including odds ratios and standardized mean differences were extracted or calculated based on studies' primary data. Comparable studies were combined by the random effects model for meta-analyses along with assessment of heterogeneity and publication bias.

### Results

A total of 36 studies were included in the final analyses. The studies were combined into different subgroups for meta-analyses as there were important differences in patient population characteristics, methodology to assess hypertension, and choice of effect size reporting (or calculability from primary data). For adults, there were no significantly increased odds of elevated blood pressures for food insecure individuals in studies where researchers measured the blood pressures: OR = 0.91 [95%CI: 0.79, 1.04; $n = 29,781$; $Q(df = 6) = 7.6$; $I^2 = 21\%$]. This remained true upon analysis of studies which adjusted for subject BMI. Similarly,

**Funding:** The authors received no specific funding for this work.

**Competing interests:** The authors have declared that no competing interests exist.

in studies for which the standardized mean difference was calculable, there was no significant difference in measured blood pressures between food secure and FIS individuals: g = 0.00 [95%CI: -0.04, 0.05; $n$ = 12,122; $Q(df$ = 4) = 3.6; $I^2$ = 0%]. As for retrospective studies that inspected medical records for diagnosis of hypertension, there were no significantly increased odds of hypertension in food insecure adults: OR = 1.11 [95%CI: 0.86, 1.42; $n$ = 2,887; $Q(df$ = 2) = 0.7; $I^2$ = 0%]. In contrast, there was a significant association between food insecurity and self-reports of previous diagnoses of hypertension: 1.46 [95%CI: 1.13, 1.88; $n$ = 127,467; $Q(df$ = 7) = 235; $I^2$ = 97%]. Only five pediatric studies were identified which together showed a significant association between FIS and hypertension: OR = 1.44 [95%CI: 1.16, 1.79; $n$ = 19,038; $Q(df$ = 4) = 5.7; $I^2$ = 30%]. However, the small number of pediatric studies were not sufficient for subgroup meta-analyses based on individual study methodologies.

## Discussion

In this systematic review and meta-analysis, an association was found between adult FIS and self-reported hypertension, but not with hypertension determined by blood pressure measurement or chart review. Further, while there is evidence of an association between FIS and hypertension among pediatric subjects, the limited number of studies precluded a deeper analysis of this association. These data highlight the need for more rigorous and longitudinal investigations of the relationship between FIS and hypertension in adult and pediatric populations.

## Introduction

Food insecurity (FIS) is defined as limited or inadequate access to food, often related to individual factors such as poverty, disability, and systemic factors like transportation or grocery store access [1]. FIS is a complex sociomedical issue which has undergone multiple definitions over time, more recently broadening from a focus on food supply alone and to include food access and quality [2]. As a result, many validated tools have been developed to assess food insecurity and its various contributing factors [3]. FIS has since emerged as a significant social determinant of health with profound and far-reaching public health consequences. In 2015, more than 42 million individuals in the US alone qualified as food insecure, a proportion greater than that found after the 2007 recession [4]. Given the importance of dietary habits in the prevention and management of chronic disease, FIS has shown many downstream effects on individual health outcomes. FIS has been associated with worse general health and increased incidences of cognitive abnormalities, birth defects and a number of chronic diseases/conditions including hypertension, diabetes, obesity, anxiety, sleep disorders, and major depression [5–7]. Due to increased disease prevalence and compromised chronic disease management, FIS has also been linked to high annual healthcare costs estimated at nearly $2,000 in added healthcare expenditure per food insecure individual per year and total national healthcare costs of almost $80 billion annually [8]. The public health importance of FIS is further highlighted by the fact that FIS is more prevalent in Black, Hispanic, and low-income households [9, 10].

At least two processes, either alone or in concert, may account for a direct or collateral link between FIS and cardiovascular disease. First, in homes with FIS, dietary habits are more likely

to involve intake of high-calorie, nutritionally-poor food, with increased concentrations of salt, dietary factors all know to increase cardiovascular risk [11–14]. In addition, it has been proposed that the "stress" of being food insecure contributes to increased chronic activation of cortisol production pathways, an important risk factor for a broad range of cardiometabolic diseases [15].

These processes have provided putative mechanistic bases for an association between FIS and cardiovascular disease and have prompted a broad range of investigations of the cardiometabolic impacts of FIS. These studies have established strong associations between FIS and cardiovascular risk factors such as glycemic intolerance, dyslipidemia, and cardiovascular events [6, 16, 17]. However, while numerous primary studies have similarly investigated the link between FIS and hypertension, no study to date has integrated this evidence. Therefore, a systematic review and meta-analysis was performed to assess the association between FIS and hypertension.

## Methods and materials

### Data sources

This review was the result of an initially broad literature search exploring the link between FIS and major cardiometabolic risk factors, as registered in Prospero as of January 28, 2020 (registration CRD42020149560). In order to conduct a thorough history of such studies, all peer-reviewed human studies of any methodology, any age group, and any publication year were included in a search of four databases: PubMed, Scopus, Embase, and Web of science. The initial search was conducted on September 9, 2019 and included all Medical Subject Headings of FIS as associated with type 2 diabetes mellitus, metabolic syndrome, dyslipidemia, or hypertension.

### Study selection

Every retrieved abstract was first randomized via Abstrackr [18] and subsequently inspected by four authors. An abstract was deemed relevant if it satisfied all of the following four criteria: (1) the study involved food insecurity or a synonymous concept, (2) the study involved either hypertension, dyslipidemia, metabolic syndrome, or T2DM, (3) the study contained quantitative primary data, and (4) the study was written in or translated to English. After this step, our focus shifted specifically to hypertension. The full text of studies whose abstracts met all the above criteria were then evaluated by three study personnel to confirm the presence of any primary data related to the connection between FIS and hypertension. All studies included in the meta-analyses were assessed for individual study bias using the AXIS tool for quality assessment of cross-sectional studies [19] which yielded no results concerning enough for the exclusion of any of the studies. Additional details and discussion regarding the results of the AXIS tool audit of study quality are described in S1 File [20–22].

### Data extraction

For each study exploring FIS and hypertension which met all inclusion criteria, data was manually extracted including: study population characteristics, study design, sample size, study measurements, and all outcomes related to hypertension and FIS. Data was then grouped by pediatric and adult populations and subsequently by type of data (i.e. odds ratios and prevalences). The effect sizes investigated in this study were the logarithm of the odd ratios (OR) and standardized mean differences (specifically Hedges' g). One of the authors first extracted the effect sizes that were directly presented in the manuscript and took note of where in each

study they were discussed. Adjusted odds ratios (AOR) were also extracted if presented by included studies. When a study did not report desired data, attempts were made to extract sufficient data to calculate effect sizes manually. For studies that divided food insecurity into different categories of severity, the primary data were pooled to calculate one effect size. If effect sizes were presented for different levels of FIS, and the primary data were not, the effect sizes were combined using the random effects (RE) model. Precise grouping of the included studies can be found in S1 File.

### Data synthesis and meta-analysis

Meta-analyses were performed if effect sizes were available in at least three comparable studies for a particular measure. Analysis was conducted using the metafor package in R, based on the RE size model [23, 24]. The RE model does not assume that the sampled populations have identical probability distributions. All results were calculated using the DerSimonian–Laird estimator [25]. For meta-analyses of ORs, the logarithm was used as the effect size. For each meta-analysis, heterogeneity across studies was calculated by first calculating the total variance (Q), the degrees of freedom (df), and the $I^2$ statistic [26]. For each meta-analysis, possible publication bias was calculated by the Begg and Mazumdar rank correlation test for funnel plot asymmetry as well as Kendall's tau in Egger's regression test [27]. When at least three studies were available which reported subgroup data, additional analyses were performed. The authors a-priori chose to explore differences in race, gender, and age as potential subgroups for analyses.

## Results

### Search results

A flow diagram of the search results and selection process of included studies is presented in Fig 1. The initial search yielded a total of 787 abstracts. The four authors that reviewed the initial selection of abstracts demonstrated perfect agreement for 84% of the abstracts. The Fleiss inter-rater kappa, a statistical measurement of the group's agreement, was calculated at 0.68 [95%CI: 0.66, 0.71, $N = 4$]. Furthermore, every user had a pair-permuted kappa above 0.6 (S2.1 Fig in S2 File) [28]. The remaining 196 abstracts were evaluated by at least two independent raters to evaluate if each abstract corresponded to a full-text, peer-reviewed article and that the manuscript presented quantitative data testing for an association between FIS and the four cardiovascular risk factors named above. For each approved full text, studies were then evaluated for the presence of primary data on the link between FIS and hypertension. Disagreements were resolved by the author who conducted the eventual statistical analysis.

### Study characteristics

Following full text evaluation, a total of 46 studies were identified which reported data examining the relationship between FIS and blood pressure or hypertension ($n = 339,007$). Ten studies were removed which were found to have either overlapping or insufficient data to calculate either an odds ratio or a standardized mean difference. The remaining 36 studies ($n = 224,766$) were all included in the final analysis. We first categorized studies into adult or children populations: 28 studies (n = 202,004) involved adults and five studies ($n = 19,038$) involved pediatric subjects and the three remaining studies ($n = 3,724$) contained data on both children and adults without reporting differentiated data and were therefore analyzed separately. The characteristics of each included study (including the population information, study design, size, blood pressure measurement method, and potential confounding information such as BMI

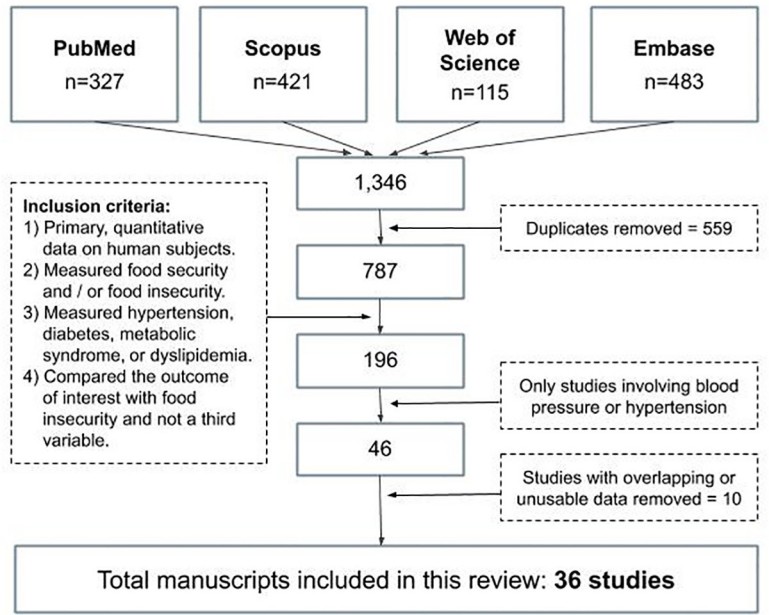

**Fig 1. Flow chart summarizing the search and selection process.** Presented is the search process followed by study selection leading to 36 included studies.

and age) are presented in the S2.1-S2.4 Tables in S2 File. Studies with sufficient data to calculate both the odds ratio and the standardized mean difference were included in multiple tables. Several subgroup meta-analyses were also conducted based on the studies' methodologies and subpopulations (S3.1-S3.5 Figs in S3 File). None of the funnel plots were significant for asymmetry (S3.6 Fig in S3 File).

## Adult studies

Among the 28 adult studies with data exploring FIS and hypertension, there were relevant differences in population characteristics and how hypertension was explored. Some studies provided single blood pressure readings measured by researchers, others provided rates of patients reporting a previous diagnosis of hypertension, and others reported rates from medical records inspection. The compiled studies also differed on whether an odds ratio or standardized mean difference were available (or calculable from primary data). We first combined the 21 adults studies ($n$ = 190,429) that contained either odds ratios or sufficient primary data to calculate odds ratios for the occurrence of hypertension (or singly-measured elevated blood pressure) between adult patients with and without FIS [29–49]. When all the studies are combined, regardless of methodology to assess hypertension, a significant combined odds ratio is obtained:1.20 [95%CI: 1.01, 1.43; n = 190,429; $Q(df$ = 20) = 426; $I^2$ = 95%]. The high heterogeneity of the results, by visual inspection and the quantified $I^2$, suggested that the 21 studies should be sub-grouped by the methodology by which hypertension diagnoses were determined. Therefore, we next explored subgroupings based on how hypertension was explored. Seven studies provided data on single-reading blood pressures performed by researchers. Fig 2 shows the meta-analysis which found lower odds of elevated blood pressure in the FIS group OR = 0.91 [95%CI: 0.79, 1.04; n = 29,781; $Q(df$ = 6) = 7.6; $I^2$ = 21%]; however the results were not significant [29, 32, 34, 39, 41, 42, 44]. In contrast, among the eight studies in which hypertension diagnoses were determined by patient self-report, meta-analysis shown in Fig 3 yielded

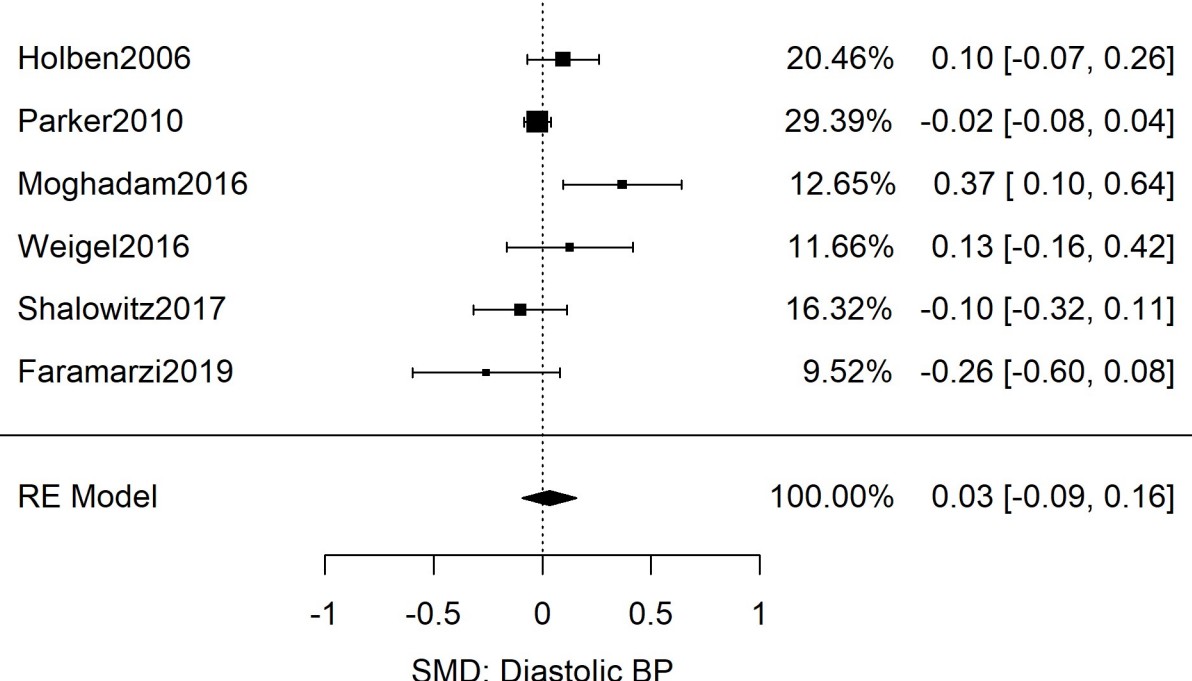

**Fig 2. Meta-analysis of adult studies investigating food insecurity and hypertension by blood pressure measurement.** Presented are the seven adult odds ratio studies involving direct blood pressure measurements, showing decreased variability in effect sizes and a non-significant combined odds ratio of 0.91 [95%CI: 0.79, 1.04; $n$ = 29,781; $Q(df = 6)$ = 7.6; $I^2$ = 21%].

a combined odds ratio of 1.46 [95%CI: 1.13, 1.88; $n$ = 127,467; $Q(df = 7)$ = 235; $I^2$ = 97%], which explained the increased OR and heterogeneity when the studies were combined [30, 31, 35, 43, 45–47, 49].

We then explored the three adult studies that inspected patient's electronic health records for hypertension diagnoses. The combined OR for these studies was 1.11 [95%CI: 0.86, 1.42; $n$ = 2,887; $Q(df = 2)$ = 0.7; $I^2$ = 0%] (S3.1 Fig in S3 File) [36, 37, 48]. The heterogeneity of these studies was below zero as it was lower than that expected from randomness in measurements. The remaining three adult studies reporting ORs used mixed methods of determining patients' hypertension and were thus analyzed separately with results reported in S3.2 Fig in S3 File [33, 38, 40].

After analyzing odd ratios as the effect size, we next explored the five adult studies ($n$ = 12,212) that contained sufficient data to calculate the standardized mean difference of systolic blood pressure between food secure and FIS patients [33, 50–53]. The results showed no difference in the systolic blood pressure readings of the food secure and food insecure groups: Meta-analysis as shown in Fig 4 yielded a combined SMD of 0.00 [95%CI: 0.04, 0.05; $n$ = 12,212; $Q(df = 4)$ = 3.6; $I^2$ = 0%]. Similarly for diastolic BP data, meta-analysis of six studies revealed a combined SMD of 0.03 [95%CI: -0.09, 0.16; $n$ = 15,240; $Q(df = 5)$ = 12.6; $I^2$ = 60%] as displayed in Fig 5 [50–55].

The number of adult studies offered the opportunity to explore further sub-group analyses based on patient characteristics and blood pressure cutoffs. We therefore performed several additional subgroup meta-analyses were conducted by grouping studies which defined hypertension using different blood pressure cutoffs as well as by grouping three studies which exclusively involved Latinx patients, none of which resulted in significant differences between food secure and FIS groups. The results of these meta-analyses are presented in S3.3-S3.5 Figs in S3

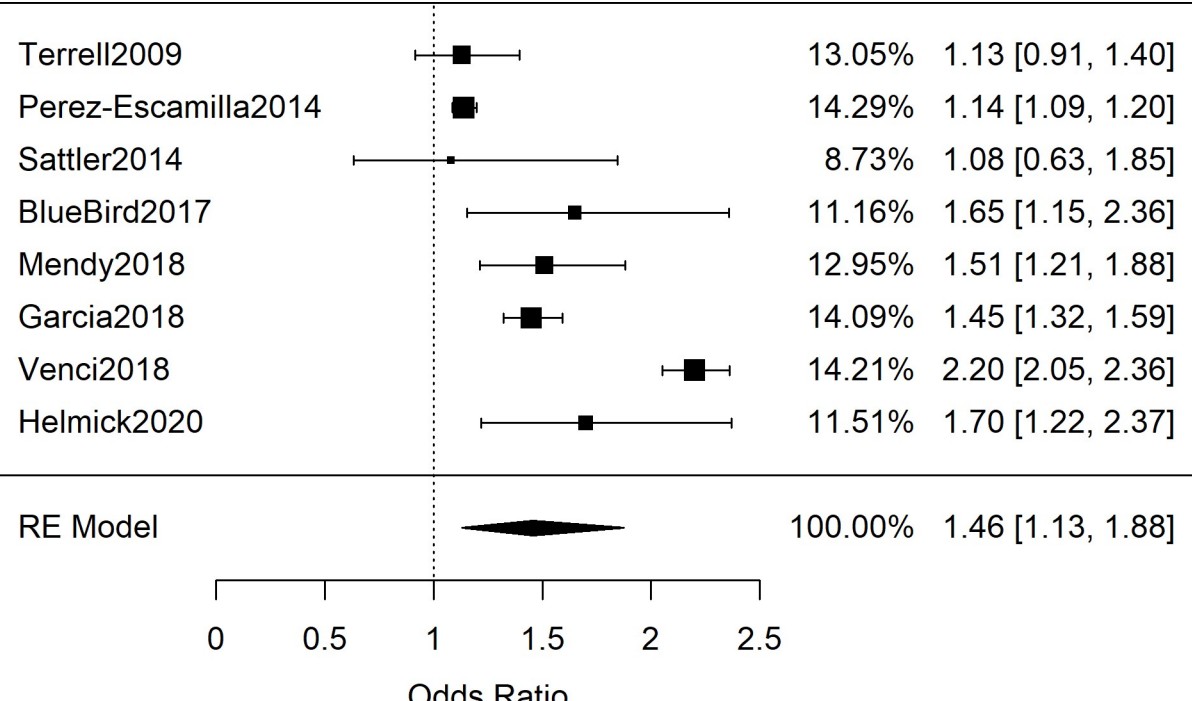

**Fig 3. Meta-analysis of adult studies investigating food insecurity and self-reported hypertension.** Presented are the adult odds ratio studies involving self-reported hypertension, demonstrating a significant, combined odds ratio of 1.46 [95%CI: 1.13, 1.88; $n = 127,467$; $Q$ $(df = 7) = 235$; $I^2 = 97\%$].

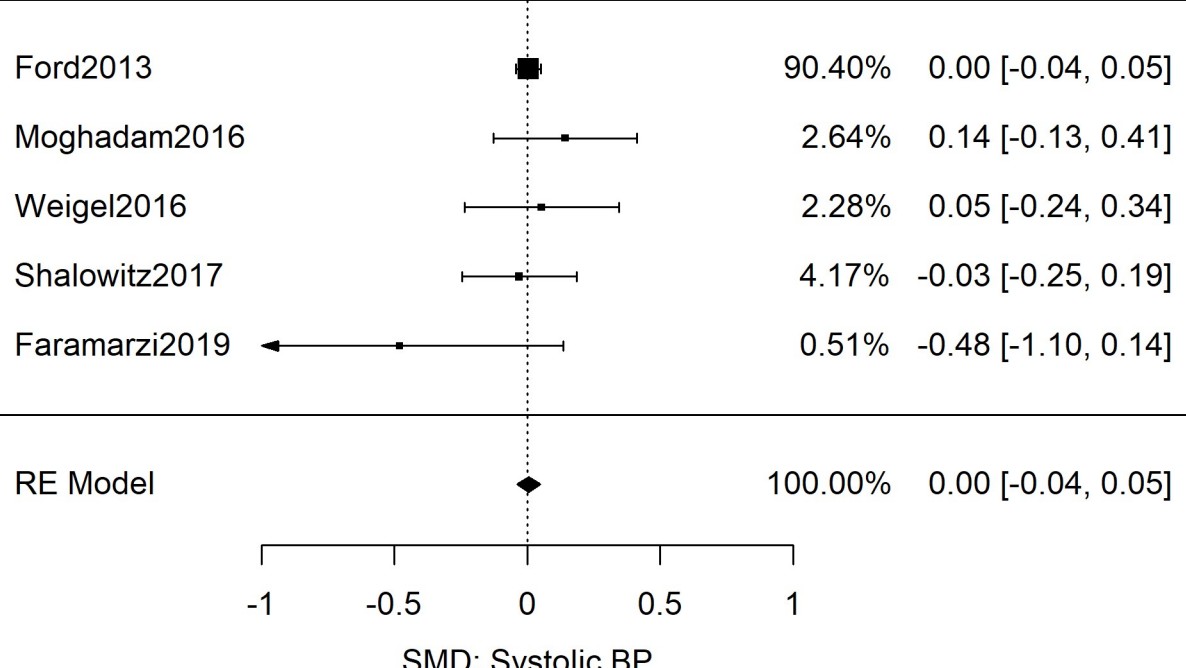

**Fig 4. Meta-analysis of SMD of food insecurity and systolic blood pressure in adults.** Shown are the results from 5 studies reporting systolic blood pressures as associated with food insecurity, resulting in a non-significant combined effect size ($g = 0.00$ [95%CI: -0.04, 0.05; $n = 12,122$; $Q(df = 4) = 3.6$; $I^2 = 0\%$]).

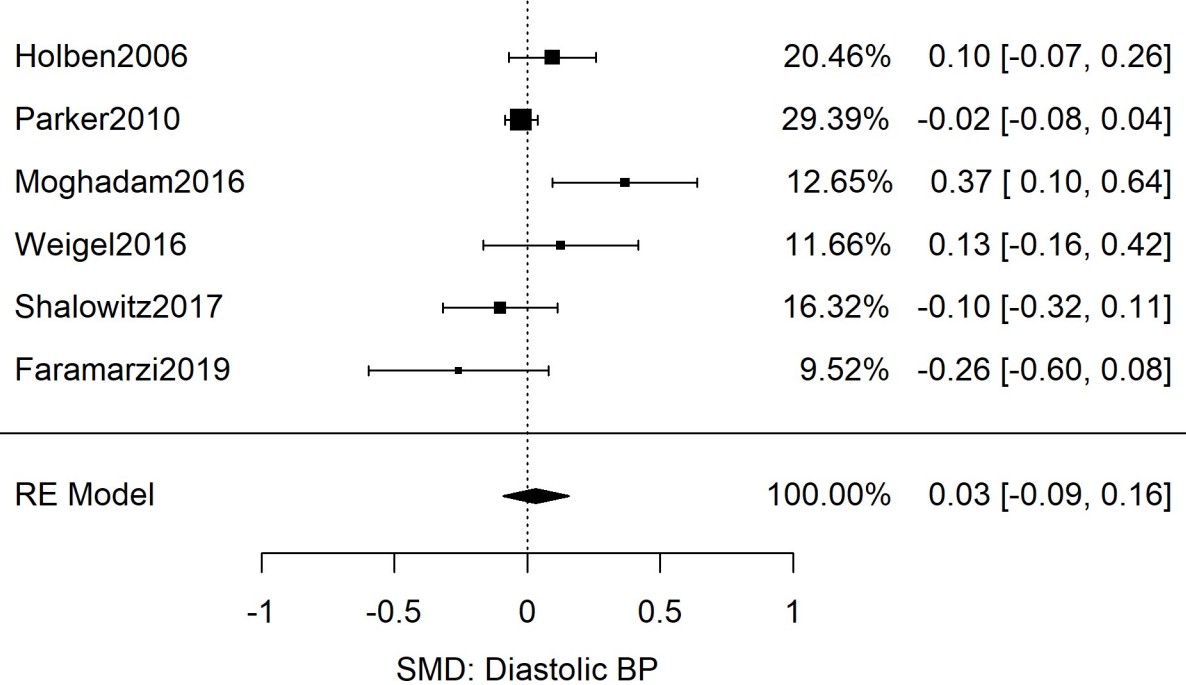

**Fig 5. Meta-analysis of SMD of food insecurity and diastolic blood pressure in adults.** Presented are the results from 6 studies reporting diastolic blood pressures as associated with food insecurity, resulting in a non-significant combined effect size ($g = 0.03$ [95%CI: -0.09, 0.16; $n = 15,240$; $Q(df = 5) = 12.6$; $I^2 = 60\%$]).

File. Due to insufficient studies, further subgrouping by sociodemographic characteristics was not attempted.

Information about covariate variables such as BMI, age, and number of medications were also extracted from the aforementioned studies as these covariates may affect the estimated association between FIS and elevated blood pressure or hypertension. As different studies provided different amount of details on these covariables, we analyzed the data in a stepwise fashion. First, we looked at the studies for which researchers measured the blood pressure. Of the seven studies, only three studies adjusted for BMI and age [32, 34, 42], and although the studies focused on different populations, the results were very similar (S2.1 Table in S2 File); the random effect model synthesized AOR was 1.02 [95%CI: 0.82, 1.27, $n = 3,257$, $Q(df = 2) = 0.22$, $I^2 = 0\%$]. The calculated heterogeneity, below that expected from random chance, was remarkable as the studies focused on different populations such as low income Malaysian women [34], diabetic adults in the USA [32], and the US Latinx population [42]. Three of the seven studies did not adjust for BMI but did report that there was no difference in BMI between the food secure and food insecure groups [29, 39, 44]; combination of the odd ratios still resulted in a non-significant association and low heterogeneity between studies: OR = 1.05 [95%CI: 0.75, 1.47, $n = 5,328$, $Q(df = 2) = 1.57$, $I^2 = 0\%$]. Lastly but importantly, the study by Berkowitz et al on the US general population was the only study of the seven for which there were differences in rates of obesity between the FIS and FS groups; the FIS group had a higher rate of obesity ($p < 0.001$) and also significant lower odds of BP (OR = 0.80 [95%CI: 0.73, 0.88]) [41]. This was the only study with a significant association between FIS and elevated blood pressure and the greatest contributor to heterogeneity in the combination of the seven studies (Fig 1). As for the three studies that inspected medical records [36, 37, 48], only one study, by Berkowitz et al, found a significant association between FIS and hypertension although it was not adjusted for subject BMI.

As for the eight studies with self-reported methodology, only one study adjusted for both age and BMI; this study on the US general population by Terrell et al. did not find a significant association between FIS and self-reported hypertension (AOR = 1.13 [95%CI: 0.91, 1.39, $n$ = 15,199] [30]. Another study evaluated the effect of FIS on the odds of self-reported hypertension at different BMI categories and found that for normal BMI the association was not significant for neither men nor women, while for obese patients the association was significant for both [31]. Compared to the two aforementioned studies, the remaining six studies contained less information to assess the effect of BMI on the association between FIS and self-reported previous diagnosis in hypertension. Lastly, three studies of varying methodology reported data on the association between food insecurity and hypertension without differentiating the data between adult and pediatric subjects (S2.4 Table in S2 File) [56–58]. The studies were not combined by meta-analyses. No adult study reported information regarding subjects' medications.

Finally, it should be mentioned that our literature review identified only one adult study, Laraia et al., that investigated the link between FIS and specifically pregnancy-induced hypertension. The study found no association at varying levels of food insecurity [59]. This study was not deemed suitable for inclusion in the other meta-analyses of this review.

## Pediatric studies

All of the five identified pediatric studies ($n$ = 19,038) either reported odds ratios for hypertension among children with and without FIS or contained sufficient data for its calculation [60–64]. Results of this meta-analysis are shown in Fig 6 and yielded a combined OR of 1.44 [95% CI: 1.16, 1.79; $n$ = 19,038; $Q(df = 4)$ = 5.7; $I^2$ = 30%]. Among these studies, two involved inspection of medical records for determining hypertension, one study involved direct blood pressure measurements, and the remaining two involved mixed methods. Therefore, due to insufficient studies, subgroup analyses among these studies by methodology or sociodemographics was not possible.

Similar to the adult studies, the influence of the covariates BMI and age was considered for the included pediatric studies. None of the five studies presented an odds ratio adjusted for BMI. However, the five studies did report that there was no statistically significant difference in BMI between FIS and non-FIS groups [60–64]. Regarding age, three of the five studies adjusted their reported AORs by subject age [60, 63, 64]; while one study reported no statistically significant difference in age between FIS and non-FIS subjects [61]. The final study did not report age-related data or specify age-adjustment in their findings [62].

## Discussion

The possibility of a positive association between FIS and hypertension is consistent with the recognized effects of calorically-dense, high salt diets, and chronic stress in raising blood pressure [11–15]. However, the impact FIS on hypertension for an individual may be complex and context-dependent as in the example of extreme FIS leading to overall lower caloric intake and salt ingestion, thereby potentially lowering individual blood pressures [65]. Nevertheless, FIS remains a significant social determinant of health warranting further investigation of the ways that FIS may affect blood pressure.

Despite several plausible mechanisms for an association between FIS and blood pressure, the results of this systematic review and meta-analysis showed no association between FIS and elevated blood pressure measurements. While the odds ratio cohorts of 21 adult studies (Fig 7; $n$ = 190,429) and 5 pediatric studies (Fig 6; $n$ = 19,038) showed a significant association between FIS and hypertension, both groups showed high variability between studies that

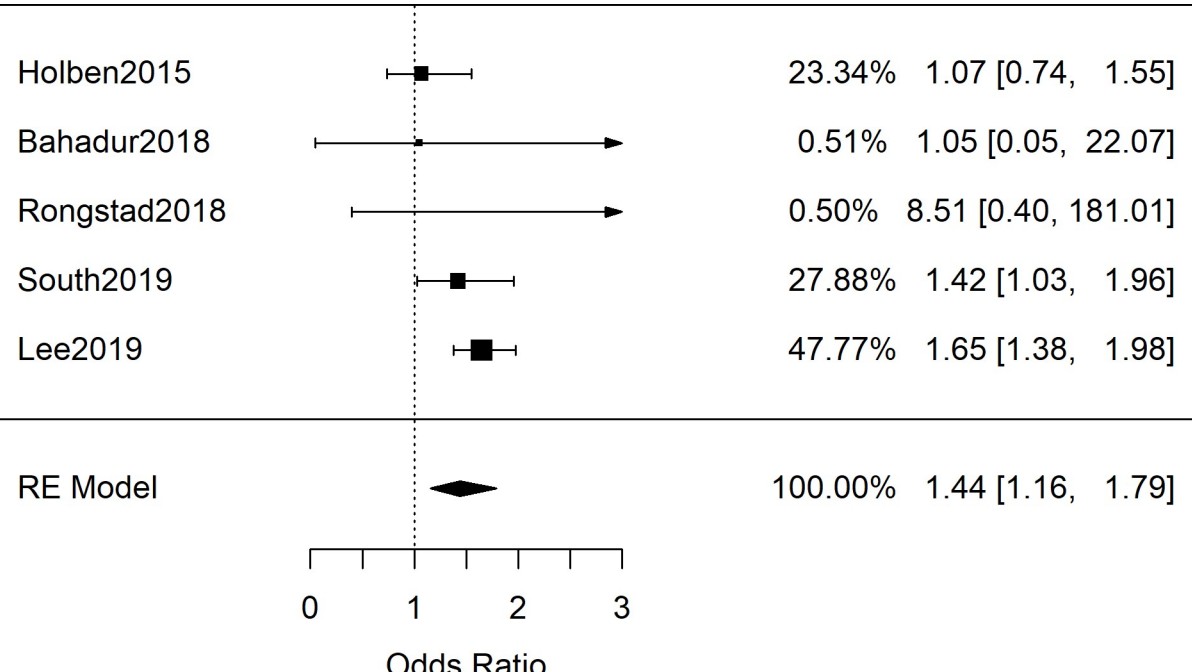

**Fig 6. Meta-analysis of pediatric studies investigating food insecurity and hypertension.** Shown are data from the 5 pediatric odds ratio studies showing variable results but a significant, combined odds ratio of 1.44 [95%CI: 1.16, 1.79; $n$ = 19,038; $Q(df = 4)$ = 5.7; $I^2$ = 30%]. Two studies involved medical records inspection of diagnosed hypertension [61, 62], two involved mixed methods for hypertension diagnoses [63, 64], and one involved direct BP measurements [60].

utilized clinical BP measurements and those with self-reported hypertension. Furthermore, the subgroup analysis revealed that the relationship persisted only among studies in which hypertension was self-reported rather than determined by clinical measurement (Figs 2 and 3). Meta-analysis of SMD for adults' blood pressure corroborated this finding for both systolic and diastolic readings (Figs 4 and 5). The results showed that FIS is specifically associated with higher self-reported hypertension, suggesting a complex relationship between FIS and hypertension.

Regarding pediatric data, this review found an insufficient number of studies in order to explore subgroup analyses between the self-reported and clinically measured data, making it difficult to discern if a similar relationship between FIS and hypertension may exist for pediatric subjects. Furthermore, due to the small number of studies, further analysis including covariates such as age or BMI was not possible. For both pediatric and adult studies, longitudinal data on the effect of FIS on future blood pressure are lacking.

There are several possible explanations for the results of this systematic review and meta-analysis. First, the finding that FIS was significantly associated with self-reported hypertension rather than a hypertension diagnosis or elevated measured blood pressure is an important finding. As others have noted, the reliability of self-reported hypertension can vary widely along patient demographic lines, with particularly high false positives found among people of low socioeconomic status [66, 67]. Furthermore, the accuracy of self-reported health depends greatly on patients' level of healthcare access and health literacy [68]. Given that low socioeconomic status is an important correlate as well as direct cause of FIS [69], issues of health literacy and a higher tendency for false positive self-reporting may, in part, explain the association found in this review.

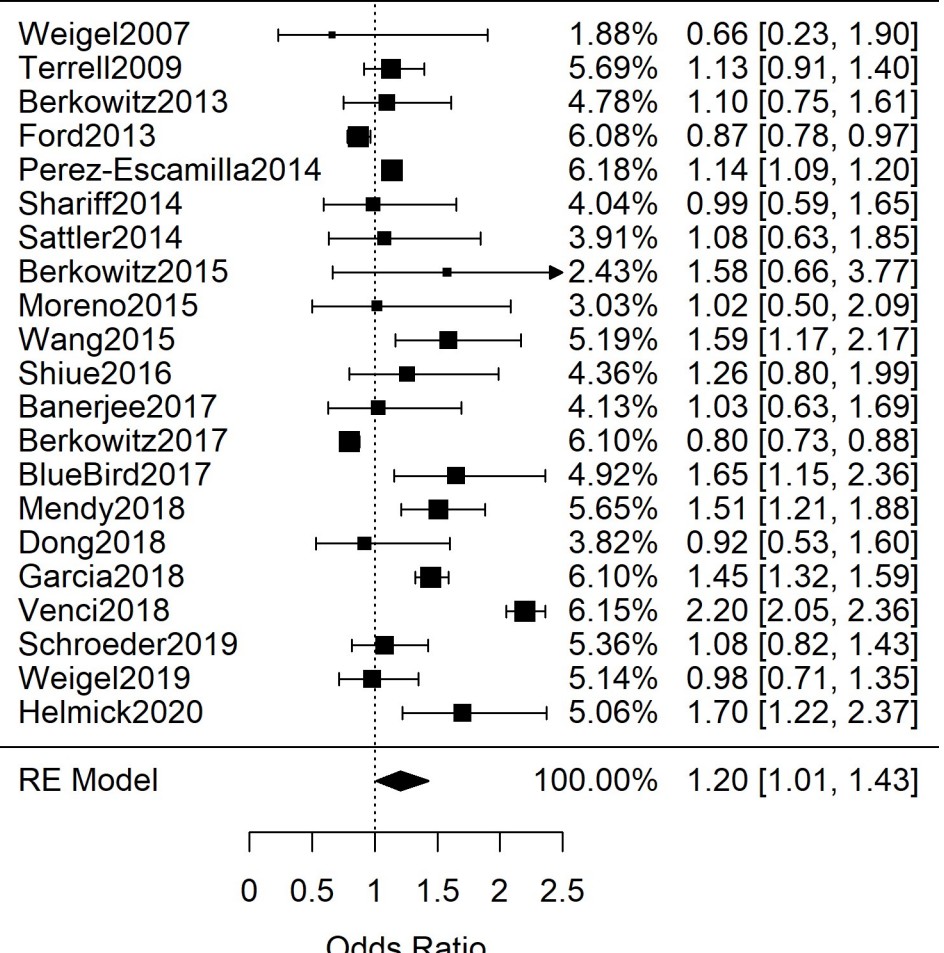

**Fig 7. Meta-analysis of adult studies investigating food insecurity and hypertension.** Shown are the 21 adult odds ratio studies investigating the association between food insecurity and hypertension demonstrating a high variability between study effect sizes and a significant, combined odds ratio of 1.20 [95%CI: 1.01, 1.43; $n$ = 190,429; $Q(df$ = 20) = 426; $I^2$ = 95%].

Another potential explanation for the association between FIS and self-reported hypertension is the role of differences in cultural conceptions of stress and anxiety. The association between FIS and anxiety has been well documented in the medical literature [7]. Furthermore, FIS and anxiety have each been found to disproportionately affect minority populations [70]. In survey or interview settings, research has shown that some minority populations may conflate the concepts of stress or anxiety with hypertension, possibly as a result of differences between the intended meaning of the clinical terms and how those terms are understood colloquially [71, 72]. For example, drawing from their own cultural perspective, a patient may interpret an interviewer's question about blood pressure to refer also to their self-perceived stress. It is possible that a higher self-reported rate of hypertension among patients with food insecurity may partially be a detection of increased self-reported stress. Such an explanation of the above findings may place further emphasis on the use of culturally-informed methods to assess self-reported chronic disease among underserved populations.

The increased variability seen in the SMD diastolic blood pressure data compared to the systolic blood pressure data was also a notable finding of this review. One explanation for this

result is the tendency for diastolic blood pressure readings to be more error-prone than systolic readings [73, 74]. Another explanation could be the natural evolution of systolic and diastolic blood pressure with age. While systolic blood pressure has a positive linear relationship with age, diastolic blood pressure begins to decrease in the fifth and sixth decade of life [75, 76]. Our SMD analysis was limited to adult studies and therefore, may have revealed a natural clustering of systolic blood pressure data with varied diastolic blood pressure data.

This systematic review and meta-analysis have several strengths. First, as our initial search included all studies in which FIS was assessed with four major cardiovascular risk factors, many more studies underwent full text evaluation for data on FIS and blood pressure or hypertension than would have otherwise been possible under a narrower search. Therefore, it is possible that more studies were identified which contained data on blood pressure or hypertension and FIS within studies whose primary focus may have been one of the three other included concepts. In fact, it is worth noting that a significant proportion of the FIS and blood pressure data included in this review was extracted from studies in which hypertension was not a primary outcome. Additionally, as this review placed no limitations on country of origin or publication year of the included studies, the analyses included a significantly larger pooled subject population than would have been possible under such restrictions.

This review also has important limitations. First, all the retrieved and combined data was derived from cross-sectional studies, making it impossible to draw conclusions regarding a possible temporal relationship between FIS and blood pressure or hypertension. Specifically, the potential for the latent action of chronic FIS on blood pressure cannot be ascertained from this data. This limitation is particularly important in the pediatric population for which there may be high variability depending on patient age or duration of FIS. Similarly, our subgroup analyses were greatly limited by the availability of demographic-specific data on FIS and hypertension. Therefore, the possibility of a group-level association between FIS and hypertension masked by pooled data remains unclear. As for publication bias, although no funnel asymmetry test was significant, it is important to note that the statistical power of these tests is small for the number of reviewed studies [27, 77, 78].

Second, the majority of studies which determined subjects' hypertension by clinical measurement used single blood pressure readings rather than the multiple blood pressure readings typically needed to confidently establish or exclude a diagnosis of hypertension, thereby presenting a significant limitation [79, 80]. However, the non-significant difference in odds for elevated BP between food insecure and food secure patients mirrors the non-significant results for the studies that inspected medical records. The medical records studies, however, do not escape the limitation that there was no information about primary or secondary hypertension. Therefore, the current peer-reviewed data is not sufficient to ascertain a possible relation between FIS and secondary hypertension, which is less prevalent than primary hypertension. Another important limitation in exploring the association between FIS and hypertension from the current literature is that many studies did not adjust for covariates such as BMI and number of hypertension medications. However, sub group analysis of studies that adjusted for age as well as BMI showed a non-significant association between FIS and blood pressure and homogeneous results. Although the existing literature suggests that BMI and FIS do not have a strong effect size in their association [6], how BMI may affect the relationship between FIS and hypertension remains of relevance and warrants further investigation. The field would likely benefit from future studies exploring the association between FIS and blood pressure or hypertension additionally making their primary data for BMI, as well as other covariates, directly available. Finally, as noted previously, for studies which used surveys or interviews, the use of patient self-reported data in measuring chronic disease prevalence presents unique challenges and major limitations given the dependence of such methodologies on terminology for which

cultural differences may carry an important influence. Therefore, conclusions based on such data should be viewed with additional skepticism.

The findings of this review lend further nuance to the proposed links between FIS and hypertension. And the detailed results should be useful in motivating and guiding future studies. First, the negative findings on the association between FIS and BP suggest that, from the mechanistic perspective, there is a need for more studies to report direct quantitative associations between FIS and estimated salt and calorie intakes. Several studies across different populations will be required to understand the granularity of a FIS-hypertension association. Second, the increase in self-reported hypertension, instead of clinically measured blood pressure, also suggests that future studies should consider favoring clinically measured data; and that psychological distress and/or anxiety are potentially confounding variables worthy of measurement if the study design permits.

## Conclusions

This study highlights the need for further exploration of both longitudinal and demographic-specific data regarding FIS and blood pressure in adults. Further, there are insufficient data to draw meaningful conclusions regarding a possible relationship between childhood FIS and elevated blood pressure as well as the possible impact of chronic childhood FIS on blood pressure into adulthood. Longitudinal studies are also needed to determine the short- term and life-long effects of chronic FIS on blood pressure in pediatric subjects.

## Supporting information

**S1 File.**
(PDF)

**S2 File.**
(PDF)

**S3 File.**
(PDF)

**S4 File.**
(PDF)

## Author Contributions

**Conceptualization:** Sourik Beltrán, Marissa Pharel, Daniel J. Arenas.

**Data curation:** Sourik Beltrán, Marissa Pharel, Canada T. Montgomery, Itzel J. López-Hinojosa, Daniel J. Arenas.

**Formal analysis:** Sourik Beltrán, Marissa Pharel, Canada T. Montgomery, Itzel J. López-Hinojosa, Daniel J. Arenas.

**Methodology:** Sourik Beltrán, Marissa Pharel, Daniel J. Arenas.

**Project administration:** Sourik Beltrán, Daniel J. Arenas.

**Resources:** Horace M. DeLisser.

**Software:** Daniel J. Arenas.

**Supervision:** Sourik Beltrán, Daniel J. Arenas.

**Writing – original draft:** Sourik Beltrán, Daniel J. Arenas.

**Writing – review & editing:** Sourik Beltrán, Marissa Pharel, Canada T. Montgomery, Itzel J. López-Hinojosa, Daniel J. Arenas, Horace M. DeLisser.

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
