## [Decision Letter · Decision Letter 0]

30 Jul 2020

PONE-D-20-15615

Food Insecurity and Hypertension: A Systematic Review and Meta-analysis

PLOS ONE

Dear Dr.DeLisser,

Thank you for submitting your manuscript to PLOS ONE. After careful consideration, we feel that it has merit but does not fully meet PLOS ONE’s publication criteria as it currently stands. Therefore, we invite you to submit a revised version of the manuscript that addresses the points raised during the review process.

Please make sure to answer all reviewers' questions. 

We look forward to receiving your revised manuscript.

Kind regards,

Ronpichai Chokesuwattanaskul, M.D.

Academic Editor

PLOS ONE

Journal Requirements:

Reviewers' comments:

Reviewer's Responses to Questions

**Comments to the Author**

1. Is the manuscript technically sound, and do the data support the conclusions?

Reviewer #1: Yes

Reviewer #2: No

2. Has the statistical analysis been performed appropriately and rigorously? 

Reviewer #1: Yes

Reviewer #2: N/A

3. Have the authors made all data underlying the findings in their manuscript fully available?

Reviewer #1: Yes

Reviewer #2: Yes

4. Is the manuscript presented in an intelligible fashion and written in standard English?

Reviewer #1: Yes

Reviewer #2: Yes

5. Review Comments to the Author

Reviewer #1: Dear editor in chief,

Thank you for inviting me to review the above-referenced paper. This research paper by Sourik Beltrán et al., aims to explore the association between food insecurity (FIS) and hypertension through a systematic review and meta-analysis. They found a significant association between FIS and self-reported hypertension in adults and in pediatric subjects. I think that the manuscript can be accepted for publication after taking into account the following minor comments:

• It should be noted that the association studied is between essential hypertension and FIS, and in this case it is necessary to eliminate any study which studies the secondary hypertension.

• The conflict of interest is not mentioned in the manuscript.

Reviewer #2: - Introduction needs to fulfill the criteria of the direct or predisposing relation between hypertension and food insecurity moreover, definition of food insecurity needs detailed illustration.

-No available data in the metanalysis regarding the stage of hypertension or number of medication for patients and food insecurity.

-Mean BMI of the group studied and association of other cardiovascular risk factors that may interfere with blood pressure control.

-In line 365 mentioned that majority of patients were subjected as hyperattentive from single reading which interfere with the basis of this metanalysis.

-In line 368 patient self-reported data in measuring chronic disease also not reliable as inclusion study.

-More specific studies should be included to achieve available recommendation and conclusion.

6. PLOS authors have the option to publish the peer review history of their article (what does this mean?). If published, this will include your full peer review and any attached files.

Reviewer #1: No

Reviewer #2: No

---

## [Author Response · Author response to Decision Letter 0]

2 Sep 2020

Responses to Reviewers

Reviewer #1

1. It should be noted that the association studied is between essential hypertension and FIS, and in this case it is necessary to eliminate any study which studies the secondary hypertension. 

We thank the reviewer for this important point. The articles identified by this systematic review differed greatly, not only in regard to patient populations, but also in the methodology of exploring the association between FIS and hypertension. Due to insufficient studies, granularity on whether FIS causes primary and/or secondary hypertension cannot be answered in this review. Rather, we chose to group studies based on methodology and outcomes rather than particular patient populations in order for meta-analysis to be possible. To better illustrate this point, we altered the results section of the abstract to better clarify the choice to group studies based on methodology (lines 66 - 69). Furthermore, we added a brief mention of the inability of this review to assess whether food insecurity is associated with specifically primary or secondary hypertension to the limitations section (lines 439 - 442).

2. The conflict of interest is not mentioned in the manuscript.

We appreciate the reviewer’s attention to this detail. A conflict of interest statement has been added to the manuscript following the conclusions section (lines 476 - 477).

Reviewer #2

1. Introduction needs to fulfill the criteria of the direct or predisposing relation between hypertension and food insecurity moreover 

We agree with the importance explicitly describing the potential mechanistic link between FIS and hypertension and in fact provided this information in the final 2 paragraphs of the Introduction.

2. [The] definition of food insecurity needs detailed illustration.

We have added additional text in lines 96 - 99 of the Introduction regarding the definition of food insecurity.

3. No available data in the metanalysis regarding the stage of hypertension or number of medication for patients and food insecurity.

We thank the reviewer for this observation. We agree that specific information regarding the staging of hypertension or the use of hypertensive medications between food insecure and food secure groups is an important consideration. Unfortunately, the studies identified in this review did not provide data related to the staging of hypertension as related to food insecurity. Furthermore, upon an additional review of the included studies, we found that no study reported data about the number of medications used by subjects. To make this point, we added this observation to the results section (lines 290 - 291 and lines 323 - 324).

4. Comment: Mean BMI of the group studied and association of other cardiovascular risk factors that may interfere with blood pressure control.

The authors agree with the reviewer’s point that BMI would be an important consideration in this study. Upon re-evaluation of the included studies, we found important details regarding which studies reported BMI information or adjusted their results for subject BMI. These additional results are detailed in the results section in lines 290 - 323. As reporting of BMI data was inconsistent across included studies, this additional analysis is also mentioned in the limitations section in lines 443 - 451. Finally, as this is an important consideration in the interpretation of our results, we also added a line to mention adjustment for BMI in the abstract (lines 71 - 72).

5. In line 365 mentioned that majority of patients were subjected as hypertensive from a single reading which interfere with the basis of this metanalysis.

We agree with the reviewer’s point that a single elevated blood pressure reading does not constitute a diagnosis of hypertension. We believe that this is an important limitation of several studies included in this comprehensive review. To more clearly illustrate this point, we have expanded the previous discussion on this point (see lines 436 - 439)

6. In line 368 patient self-reported data in measuring chronic disease also not reliable as inclusion study.

Similar to the previous point, we agree with the reviewer’s observation regarding the unreliability of self-reported hypertension. This is an indeed a notable limitation of some of the included studies. To further clarify this point, we have added additional language in lines 452 - 456 of the limitations section.

7. More specific studies should be included to achieve available recommendation and conclusion.

We appreciate the reviewer’s concern with making recommendations or conclusions based on the results of our study. As we conducted a broad initial search of the literature, it is likely that our review encompasses almost all if not all published data on the possible relationship between food insecurity and hypertension or elevated blood pressure. We have clarified this point in the abstract section, lines 55 - 56. As such, we are unable to include more specific studies which could provide a greater understanding regarding food insecurity and blood pressure or hypertension. Our review found a largely negative association between food insecurity and hypertension or elevated blood pressure suggesting a need for further investigation into a possible relationship. As such, there are sufficient information to generate reliable recommendations at this time.

---

## [Decision Letter · Decision Letter 1]

19 Oct 2020

Food Insecurity and Hypertension: A Systematic Review and Meta-analysis

PONE-D-20-15615R1

Dear Dr. DeLisser,

We’re pleased to inform you that your manuscript has been judged scientifically suitable for publication and will be formally accepted for publication once it meets all outstanding technical requirements.

Kind regards,

Ronpichai Chokesuwattanaskul, M.D.

Academic Editor

PLOS ONE

Reviewers' comments:

Reviewer's Responses to Questions

**Comments to the Author**

1. If the authors have adequately addressed your comments raised in a previous round of review and you feel that this manuscript is now acceptable for publication, you may indicate that here to bypass the “Comments to the Author” section, enter your conflict of interest statement in the “Confidential to Editor” section, and submit your "Accept" recommendation.

Reviewer #2: All comments have been addressed

2. Is the manuscript technically sound, and do the data support the conclusions?

Reviewer #2: Partly

3. Has the statistical analysis been performed appropriately and rigorously? 

Reviewer #2: Yes

4. Have the authors made all data underlying the findings in their manuscript fully available?

Reviewer #2: Yes

5. Is the manuscript presented in an intelligible fashion and written in standard English?

Reviewer #2: Yes

6. Review Comments to the Author

Reviewer #2: regarding the manuscript PONE-D-20-15615R1

Food Insecurity and Hypertension: A Systematic Review and Meta-analysis the author responses for the previous comments as appropriate therefore, i accept this manuscript .

7. PLOS authors have the option to publish the peer review history of their article (what does this mean?). If published, this will include your full peer review and any attached files.

Reviewer #2: No

---

## [Editor Report · Acceptance letter]

26 Oct 2020

PONE-D-20-15615R1 

Food Insecurity and Hypertension: A Systematic Review and Meta-analysis 

Dear Dr. DeLisser:

I'm pleased to inform you that your manuscript has been deemed suitable for publication in PLOS ONE. Congratulations! Your manuscript is now with our production department. 

Kind regards, 

on behalf of

Dr. Ronpichai Chokesuwattanaskul 

Academic Editor

PLOS ONE